# Influence of Selected Sociodemographic and Cultural Factors on the Age of Sexual Initiation of Students from Polish Universities

**DOI:** 10.3390/ijerph20043468

**Published:** 2023-02-16

**Authors:** Maciej Stokłosa, Iga Stokłosa, Gniewko Więckiewicz, Mateusz Porwolik, Maciej Bugajski, Tomasz Męcik-Kronenberg, Robert Pudlo, Piotr Gorczyca, Magdalena Piegza

**Affiliations:** 1Department and Clinic of Psychiatry, Medical University of Silesia, 42-612 Tarnowskie Góry, Poland; 2Department of Ophthalmology, Medical University of Silesia, University Clinical Center, 40-514 Katowice, Poland; 3Department of Pathology, Medical University of Silesia, 41-800 Zabrze, Poland

**Keywords:** sexual initiation, students, sexual education

## Abstract

Sexuality is the one of the most important parts of human life. The aim of our study was to identify the factors influencing the onset and age of sexual initiation in students and drawing attention to the need to improve access to sexual education in Polish schools at a sufficiently high level. An original questionnaire with 31 questions was used for the study. Data were collected using the Google Forms tool. A total of 7528 students participated in the study and 5824 underwent sexual initiation. The mean age at sexual initiation was 18.1 years. Logistic regression analysis was performed to show factors influencing the onset of sexual activity, while linear regression analysis was used for factors influencing the age of sexual initiation. Religion, drug use, smoking, alcohol consumption, type of housing, and conversations with parents about contraception or sex affect the onset of sexual activity. The age of sexual initiation is affected by religion, the age of the first viewing of pornography, quality of life, the size of the city of residence, smoking and drug use.

## 1. Introduction

Sexuality remains an important area in human life, and it affects the quality of life and is part of psychophysical well-being. There are many health benefits of sexual intercourse, including better mental and physical health indicators in people who have vaginal intercourse [1]. Studies have shown that earlier sexual intercourse can affect sexual health (including sexually transmitted infections or adverse sexual events), but on the other hand, it also has a positive effect on sexual function, such as less pain during vaginal penetration, better orgasmic function, and lower sexual inhibition [2]. In a study of Jamaican adolescents, the age of sexual initiation was found to be about 14 years for one-third of the respondents, well below the age at which they have access to adequate sex education. Introducing appropriate sex education in this area could help delay sexual intercourse and reduce factors that promote risky sexual activity [3]. Factors described in other works that may influence girls’ earlier sexual initiation include low social and economic as well as cultural status, low parental restrictions, parental separation and lack of religiosity [4]. It has been demonstrated that sexually active individuals during the COVID-19 pandemic had significantly lower rates of depression and anxiety than those who remained abstinent [5]. The age of sexual initiation varies according to sociocultural conditions. In Poland, the average age of sexual intercourse is approximately 18.8 years for boys and 16.6 years for girls, whereas in Germany, for example, the age of sexual initiation is approximately 14.5 years [6]. Over the years, and as cultural trends and social conditions change, the age of sexual initiation is also decreasing in Poland. In a survey of the inhabitants of one Polish voivodeship, it was found that the age of sexual initiation deviates statistically significantly in recent decades—from an average age of 18.97 years among those born in 1970–1975 to 16.41 years among those born in 1991–1995 [7]. A 1993 cohort study showed that early age of sexual initiation, i.e., under 16 years of age, may contribute to the development of depressive disorders in adolescents [8]. Earlier initiation of sexual activity has also been associated with more sexual partners, prevalence of sexually transmitted diseases (STDs), and nonuse of condoms [9]. Previous analyzes show that many factors such as place of origin, marital status, and education have an impact on age of sexual initiation [10]. The early age of sexual intercourse is also influenced by race; according to studies in the United States, African American men and people of Hispanic origin have sexual intercourse at the earliest age, while Caucasians choose sexual intercourse later [11]. Studies comparing the age of sexual initiation between whites and blacks have shown that blacks have a significantly lower age of first sexual contact, hypothesizing that this is related to centralization and concentration, which may also affect adverse political, economic and social conditions. In contrast, studies of individual family and social factors have not fully explained racial differences between the ages of those who have sexual intercourse [12]. Research has shown that age at initiation of sexual intercourse varies substantially across ethnic groups in some multiethnic societies, possibly due to sexuality as defined in communities and cultures, as well as social expectations, but these factors have not been adequately explored. Another study has shown that African Americans tend to have sex more quickly, which may be related to poorer access to sex education and lack of knowledge in the area of sexuality, while Asians tend to delay sex, which may be related to the belief that premarital sex can bring shame to the family [11]. In Poland, access to sexuality education is still very limited, and there are no classes in schools that introduce young adults to the mysteries of sexual life and how to make informed and safe decisions regarding health and sexuality. In a study conducted by Zbigniew Izdebski in 2022, it was found that about 30% of respondents said that sex education in Poland did not take place at all or was of insufficient quality. The problems with sex education in Poland may be related to the fact that issues related to sexuality continue to be the subject of political and ideological contention, both in terms of the form of delivery and the selection of appropriate textbooks, as well as pressure to promote content consistent with religious teachings, conservatism and a heteronormative model of relationships, while doubling down on prejudice against representatives of other sexual orientations. The teaching implemented in Poland deviates from the standards stated by the WHO, which is a violation of the WHO’s Universal Declaration of Sexual Rights [13]. Awareness of the need to introduce sex education in schools in our country should be increased and access to knowledge in this area should be ensured for all, especially given that there is scientific evidence that such teaching helps to increase sexual health knowledge and strengthen positive beliefs about contraceptive use [14]. In addition, the risk of teenage pregnancy is substantially lower among individuals who receive comprehensive sex education than among adolescents who do not have access to sex education or whose education is based only on promoting sexual abstinence as the most effective method of protecting against unintended pregnancies [15]. The aim of our study was to find out which factors have an influence on the onset and age of sexual initiation. We also wanted to draw attention to the problem of sex education and the growing demand for this topic in Polish schools, as evidenced by students’ opinions about the need to address issues of sexual life both at school with qualified teachers and in conversations with parents.

## 2. Materials and Methods

### 2.1. The Questionnaire and Its Distribution

This study is part of a larger project on the sexuality of Polish college students. The first part of the study on risky sexual behavior was published in February 2021 [16]. An original questionnaire was used for the study, which consisted of 31 questions about demographic data, drug use, sexual lifestyle habits, attitudes toward sex education and religious beliefs. Of the 31 questions, 26 were single choice and 5 were multiple choice. In this study, we analyzed 20 of the 31 questions that were previously included in a larger database (Appendix A). The original questionnaire was reviewed by two independent experts in the field of sexology, dermatology and venereology. The survey was initially administered to a group of 30 volunteers—students from the Medical University of Silesia in Katowice. In order to make it as convenient as possible for the respondents to fill in the survey and to maintain the highest possible anonymity, the Internet method was used with the Google Forms tool. The 50 largest Polish universities were randomly selected. Then, the administrators of the groups uniting students of certain years and fields of study were contacted via social media (Facebook). Each time, the administrator’s consent was obtained to disseminate the survey in an Internet group. Students agreed to participate in the study and were informed of its purpose. In the case of research conducted via online surveys, the consent of the Bioethics Committee of the Medical College of Silesia is not required. The questionnaires were collected between 12 March 2016 and 12 April 2016. Our research is exploratory and its aim was to draw attention to the factors that influence the delay of sexual initiation among high school students, with particular attention paid to the possible influence of sex education on the proper development of adolescent sexuality.

### 2.2. Sample Characteristics

A total of 7528 students (71% women and 29% men) aged 18–26 years participated in the study. The criterion for dividing respondents into two groups was the fact that sexual initiation was noted in the questionnaire. Students indicated the age at which they had their first sexual intercourse.

### 2.3. Statistics

The STATISTICA 13.3 program (StatSoft, Cracow, Poland) was used for data analysis. The Chi-square test was used to compare the qualitative variables, whereas the Mann–Whitney U test was used for the quantitative variables. Logistic regression analysis was performed to show parameters influencing the binominal variable—the fact of the onset of sexual activity—while linear regression analysis was used to determine the factors influencing the quantitive variable—the age of sexual initiation. The significance level was set at *p* < 0.05.

## 3. Results

### 3.1. Sociodemographic Factors

Table 1 shows the characteristics of the sociodemographic factors of the population studied.

Of the 7528 respondents, 1704 had not experienced sexual initiation. A comparative analysis was performed between the group that had not experienced sexual initiation and the group that had. There was no statistically significant difference between the groups in terms of sex structure. A significantly higher age was found in the group of students after sexual initiation.

The mean age at sexual initiation was 18.1 years, 17.9 years for males and 18.2 years for females. There is a statistically significant difference between groups in terms of place of origin—sexual initiation occurred latest in rural areas and earliest in large cities. In the group of individuals who did not undergo sexual initiation, there was a significantly higher proportion of individuals who lived less independently (with their family or in a dormitory—54% vs. 43% overall), while the proportion of individuals who lived alone (56% vs. 46%) was higher among those who underwent sexual initiation. There was a significant difference in the distribution of students in relation to field of study in both groups, being most evident in the lower sexual activity of students majoring in engineering and technology. There was no significant difference between groups in terms of reported social conditions as well as in the education of the respondents’ parents.

### 3.2. Sexual Behaviour and Stimulants

Table 2 presents data on sexual behavior, drugs and other variables analyzed.

Students who did not undergo sexual initiation were much less likely to use all stimulants—cigarettes, alcohol and drugs. Non-heteronormative respondents were significantly more likely to undergo sexual initiation than heteronormative respondents. Religious students are less likely to start their sexual life. Individuals who have experienced sexual initiation are significantly more likely to believe there is a need for sex education in schools.

Respondents who have not experienced sexual initiation are significantly less likely to masturbate and less likely to watch pornography. Similarly, a larger group of sexually inactive students have never masturbated or watched pornography, while respondents who have not experienced sexual initiation were significantly late in watching pornographic material for the first time. Students who are not sexually active are significantly more likely to admit that they have never talked to their parents about issues related to sexuality, STDs and contraception, and they are much more likely to have talked about contraception than the group that underwent sexual initiation; the difference in talking about sex is borderline statistically significant and there is no significant difference in talking about STDs.

### 3.3. Age of and Reason for Sexual Onset

Detailed data on reason for and age of sexual onset can be found in Table 3. Men were sexually initiated significantly earlier than women. The age of sexual initiation between genders was analyzed in detail and the earlier age of initiation in males is observed up to the 10th percentile of the study population. From the 15th percentile to the 97th percentile, the age of initiation is the same, and unlike the 99th percentile, the age of initiation is later in males.

There is statistical significance between the sexes in terms of the reason for initiating sexual initiation; men initiate more often than women out of curiosity and to satisfy their libido, while women initiate more often out of love.

### 3.4. Statistical Analysis of the Factors Affecting the Onset of Sexual Activity (Logistic Regression Analysis)

As shown in Table 4, the logistic regression analysis found that from all factors, the highest odds ratio is observed for respondents for whom the Roman Catholic faith has an influence on sexual life decisions. Moreover, significant factors affecting the onset of sexual activity were found to be drug use, alcohol consumption, living in student housing, not living with parents, talking with parents about contraception or sex and gender.

### 3.5. Statistical Analysis of the Factors Affecting the Age of Sexual Initiation (Linear Regression Analysis)

#### 3.5.1. Univariate Analysis of the Factors Affecting the Age of Sexual Initiation

The results of univariate analysis presented in Table 5 showed that the age of sexual initiation differs in smoking, alcohol consumption, drug use, place of residence, frequency of masturbation, frequency of pornography viewing, age of first pornography viewing and religiosity (*p* < 0.05).

#### 3.5.2. Multivariable Analysis of the Factors Affecting the Age of Sexual Initiation

The age of sexual initiation is a dependent variable. Smoking, religiosity, age of first pornography viewing, life conditions, the size of the city of residence, alcohol drinking, frequency of masturbation, frequency of pornography viewing and drug use were set as independent variables with *p* < 0.15 after univariate analysis. The inclusion and exclusion criteria were set at 0.05. The independent variable values are presented in Table 6.

#### 3.5.3. Multivariable Regression Result

Table 7 shows that six statistically significant factors entered the regression equation. According to the standard partial regression coefficient, the factors affecting the age of sexual initiation were smoking, religiosity, age of first pornography viewing, the size of the city of residence, life conditions and drug use.

## 4. Discussion

According to a study conducted by Zbigniew Izdebski in 2006, the average age of sexual initiation was set at 18.08 years for men and 18.82 years for women [17]. Analyzing previously conducted studies on similar topics in relation to the age of sexual initiation, there is a clear tendency to initiate sexual intercourse with subsequent generations at an earlier age [18]. Studies conducted over a similar time period in 17 European countries (the Polish population is not included in this summary) have shown that the average age of sexual initiation is approximately 16.3 years and varies between geographic regions from 15.9 years in Western countries (Austria, Belgium, France, Germany, the Netherlands and Switzerland) to 16.9 years in Eastern countries (i.e., Bulgaria, Hungary, Russia and Ukraine) [19]. It has been concluded that differences in the age of sexual initiation are influenced by demographic factors, health care systems and religiosity at the national level.

Significant differences between the sexes were found, indicating that women decide to have sexual intercourse later than men. This tendency has also been confirmed in studies conducted in recent years. It was repeatedly found that men have sexual intercourse significantly earlier than women [20,21].

Religiosity is the factor that has the strongest influence on the lack of sexual initiation in the group of respondents. Similar trends have been shown in other studies, which have repeatedly demonstrated a strong association between religious practices and religious affiliation and delayed sexual intercourse [22]. In our previously published study conducted with the same group of respondents, religious affiliation was also shown to influence the lower incidence of risky sexual behavior, such as sexual intercourse with a random person without protection or accidental sexual intercourse under the influence of alcohol, which has also been confirmed in other studies [16,23].

Another verified factor in the respondents’ answers is the place of residence. Those who delay sexual intercourse live in villages and small towns, as opposed to those who live in large cities. A study conducted in Ethiopia found different trends. There, an earlier age of sexual initiation was associated with rural residential areas compared to urban and more urbanized areas [10]. It is reasonable to assume that, depending on the country and cultural context, residence in certain areas is associated with better access to sex education and public health services, which may influence important sexual life decisions [24]. With regard to our study, it is worth noting that in the Polish population, people living in villages are more likely to be religious, and religion may be a factor determining sexual life [25].

Taking into account the housing of Polish college students, it turns out that people living with their parents or in dormitories delay sexual initiation compared to those living alone. Similarly, a study on the Chinese population has shown that living with both biological parents influences the onset of sexual activity at a later age [26]. Other studies have also shown that parental guidance reduces the probability of early sexual initiation, especially in girls [27].

This study shows that in the Polish population, parental education has no influence on the age of sexual initiation. Different results were obtained by a study from Nigeria, which showed that the age of sexual initiation among adolescents increases with the higher education of parents. A possible reason for such a relationship could be the fact that in some countries parents with a higher education are more likely to openly address the topic of sexuality at home and educate adolescents in the area of sexual intercourse than parents with lower levels of education [28].

In determining the influence of the educational sector on the initiation of sexual activity, it is noted that people who study technical and engineering subjects have sexual intercourse to a lesser extent and decide to do so later than people who study arts or the military field. Individuals who engage in broadly understood arts and exhibit higher levels of creativity are more likely to be extroverts and more open to new experiences [29]. Such character traits may predispose people to engage in sexual behavior more quickly. A study conducted in the United States concluded that people who study in the military field are much more likely to engage in risky behavior. Professionally active people in the military field also had wider sexual histories and were more likely to engage in unprotected sexual relationships [30]. The tendency toward earlier sexual initiation among individuals associated with the military field was also confirmed in our study.

Another aspect discussed in this study is the socioeconomic status and living conditions of the respondents. In contrast to other studies using a population of Polish students, material status has no influence on the initiation of sexual activity, while studies conducted in recent years have concluded that people from poorer families or in worse social circumstances have a greater propensity to engage in sexual activity at an earlier age. This trend has been linked to lack of access to sexual education and inadequate knowledge about STDs and contraception [31].

Individuals who underwent sexual initiation were significantly more likely to use alcohol, cigarettes or psychoactive substances. Among young adults who delay their first sexual intercourse, there is a lower tendency to use stimulants. A 2013 meta-analysis showed that people who are sexually initiated at an early age are more likely to use alcohol in the future [32]. According to researchers from Thailand, earlier initiation of sexual intercourse leads to more frequent use of various types of psychoactive substances such as methamphetamine [33].

Viewing pornographic material is associated with prior sexual initiation among participants in our study. Similar trends were observed among students from Ethiopia, where sexual debut at a younger age was significantly correlated with access to pornography. This is likely related to a willingness to experiment and experience sex after viewing erotic material, the content of which has an impact on stimulating desire [34].

Previous studies also considered factors that were not analyzed in our study, such as sexual harassment, which may significantly affect the onset of sexual activity at an earlier age. Another study found that parental behavior (separation, remaining in a stable household with only one parent, living together in a cohabitation or civil partnership) may also influence the age of onset of sexual activity. A systematic review also found that peer pressure may influence early sexual intercourse, especially in girls [35,36].

As reasons for having sex, most people cited love, followed by curiosity and the desire to satisfy their sex drive. In this respect, people in Polish society do not differ from young adults in other countries, because in other studies dealing with sexuality and related aspects, it was found that emotional attachment is the factor that most strongly influences sexual initiation [37]. It seems interesting to know that men are much more likely to report having sexual contact for the first time in order to satisfy their libido. The male group of the study participants gave instinctive motives as the reason for sexual initiation almost as often as curiosity, while among females, satisfaction of sex drive was among the least frequently chosen factors. In a study conducted in Nigeria, drive satisfaction also proved to be an important factor in the male sex, but there it was found that this motive was mainly taken up by adolescents and young adults who had already had sexual experiences [38]. On the other hand, a study conducted in 2003 found that men perform sexual intercourse more often for pleasure and libido satisfaction than for higher feelings [39].

In our study, individuals who delay the onset of sexual initiation are much less supportive of the introduction of professional sex education in schools, comparing the results of this study with a study conducted in 2017 among high school students from Lithuania, which found that students searching for information about sexual life more often had their coitarche sober than those not searching for information; moreover, students whose parents had discussed sex with them were more likely to plan coitarche [40]. Participants of our survey who are much less supportive of the introduction of professional sex education in schools are also less likely to talk to their parents about sex. Unfortunately, among many families, sexuality and proper sex education is still taboo. Because sexuality is an integral part of human life and development, it is important that young adults are properly educated about sexual behavior to avoid disappointment, experiences of sexual violence or experience dissatisfaction with their sex lives in the future. The obvious benefits of sex education have been demonstrated in numerous studies, which have found that properly delivered sex education is associated with more progressive attitudes toward women, less conformity to the image of hegemonic masculinity, positive attitudes toward sexual minorities and better knowledge about sexual health, including contraception, condom use and HIV/AIDS [41,42].

## 5. Limitations

Our study has limitations. Although the questionnaires we used were evaluated by experts in the field of sexology, dermatology and venereology, they have not been validated. In addition, the results of our study cannot be applied to the cross-sectional structure of Polish society, because the population we analyzed consisted of students, i.e., a small part of Polish society. Therefore, in the future, we plan to extend the research on sexual initiation among young adults to people without schooling, also using validated questionnaires, and to assess certain personality traits and other variables influencing sexual intercourse. The data we present are from 2016, but there are no similar studies for Poland. Moreover, Poland remains a country where access to sex education is significantly limited, and in our opinion, all data on this topic are urgently needed. They can also serve as a guide for other countries when it comes to the need for access to reliable sex education. In our study, we did not analyze all the factors that may affect the age at which sexual activity is initiated. It would therefore be worthwhile to expand the research to include issues such as parental separation, childhood abuse, peer pressure and the influence of social media. Women predominated in the sample, and only college students were included in the study. Therefore, it seems important to deepen the study by analyzing the factors that influence sexual initiation, even in individuals who are not in education.

## 6. Conclusions

The delay of sexual initiation is influenced the most by religiosity, further by later viewing of pornographic material, smaller city of residence and better life conditions. There is a visible relationship between the earlier age of sexual initiation and the consumption of various stimulants (cigarettes, alcohol, psychoactive substances). Most individuals who abstain from sexual activity are educated in technology and engineering and the humanities, whereas the fewest individuals who do not engage in sexual activity are from those engaged in the arts or the military. We found similar conclusions for the onset of sexual activity while, additionally, a parameter that affects sexual debut was the type of accommodation and conversations with parents about contraception or sex.

No significant impact was found in parental education, frequency of masturbation or pornography watching, age of first STD or contraception education on age of sexual initiation or fact of sexual debut. Knowledge about contraception, conscious sexual decisions and risky sexual behavior in Poland should be constantly increased, and for which it is necessary to introduce sex education classes for adolescents and young adults.

## Figures and Tables

**Table 1 ijerph-20-03468-t001:** Characteristics of sociodemographic factors in the analyzed groups.

		ALL	Underwent Sexual Initiation		χ^2^/Z	*p* Value
			NO	YES		
Amount		7528	1704	5824		
%		100	23	77		
Mean age [years]		21.8	21.2	22.1	−16	*p* < 0.001 **
Gender	Female	71%	69%	71%	-	NS
	Male	29%	31%	29%		
Place of residence	Village	25%	30%	23%	42	*p* < 0.001 *
	<100 k inhabitants	35%	35%	35%		
	100–200 k inhabitants	10%	9%	10%		
	200–500 k inhabitants	13%	10%	13%		
	over 500 k inhabitants	18%	16%	19%		
Branch of study	Technology and Engineering	20%	24%	19%	29	*p* < 0.001 *
	Humanistic	20%	19%	20%		
	Medical	15%	15%	15%		
	Economy and administration	14%	12%	14%		
	Paramedical	11%	10%	11%		
	Other	9%	9%	9%		
	Natural science	7%	8%	7%		
	Artistic	3%	2%	3%		
	Military	1%	1%	1%		
Where do you live?	With family	31%	34%	31%	64	*p* < 0.001 *
	In student housing	14%	20%	12%		
	Independently	56%	46%	56%		

* *p* value for Chi-square test; ** *p* value for Mann–Whitney U Test; NS—non significant.

**Table 2 ijerph-20-03468-t002:** Characteristics of behaviour having an impact on sexual initiation.

		ALL	Underwent Sexual Initiation		χ^2^/Z	*p* Value
			NO	YES		
Amount		7528	1704	5824		
%		100	23	77		
Have you ever smoked cigarettes?	Never	66%	82%	62%	262	*p* < 0.001 *
	Yes	34%	18%	38%		
Have you ever drunk alcohol?	Never	18%	27%	15%	140	*p* < 0.001 *
	Yes	82%	73%	85%		
Have you ever used drugs?	Never	89%	96%	86%	156	*p* < 0.001 *
	Yes	11%	4%	14%		
What is your sexual orientation?	Heterosexual	88%	92%	87%	32	*p* < 0.001 *
	Bisexual	7%	6%	8%		
	Homosexual	4%	2%	5%		
	Other	0%	0%	0%		
Do you masturbate?	Never	15%	26%	12%	184	*p* < 0.001 *
	Only in the past	11%	12%	10%		
	Less than once a week	44%	34%	47%		
	More than once a week	29%	28%	31%		
Do you watch pornography?	Never	21%	32%	18%	146	*p* < 0.001 *
	Only in the past	16%	17%	16%		
	Less than once a week	46%	36%	49%		
	More than once a week	17%	16%	18%		
At what age did you first watch a pornographic movie?		14.45	14.69	14.39	2.8	*p* < 0.005 **
Have you raised the following subjects with your parents?	Sex: Yes	35%	33%	36%	32	*p* = 0.026 *
	Contraception: Yes	40%	31%	43%	75	*p* < 0.001 *
	STDs: Yes	17%	15%	17%	-	NS
	None: Yes	53%	59%	51%	42	*p* < 0.001 *
Do your religion beliefs influence your sex life?	No	66%	43%	73%		*p* < 0.001 *
	Yes	34%	57%	27%	532	
Do you think there should be sex education in school?	No	5%	9%	3%	113	*p* < 0.001 *
	I do not have an opinion	6%	10%	5%		
	Yes	10%	80%	91%		

* *p* value for Chi-square test; ** *p* value for Mann–Whitney U Test; NS—non significant.

**Table 3 ijerph-20-03468-t003:** Characteristics of the sexual initiation of the studied population—age of and reason for sexual onset (pc—percentile).

		Underwent Sexual Initiation: YES			χ^2^/Z	*p* Value
		All	Male	Female		
Amount		5964	1708	4256		
%		78	29	71		
Age of sexual initiation [years]	1 ’ pc	13	12	14	394	*p* = 0.0017 **
	3 ’ pc	15	14	15		
	5 ’ pc	15	14	15		
	10 ’ pc	16	15	16		
	15 ’ pc	16	16	16		
	Q1	17	17	17		
	Median	18	18	18		
	Mean	18.12 ± 2.09	17.96 ± 2.24	18.18 ± 2.01		
	Q3	19	19	19		
	85 ’ pc	20	20	20		
	90 ’ pc	21	21	21		
	95 ’ pc	22	22	22		
	97 ’ pc	22	22	22		
	99 ’ pc	23	24	23		
Why have you decided to start your sex life? [%]	Love	63	49	69	3.1	*p* < 0.001 *
	Curiosity	20	24	18		
	To fulfil needs	10	21	6		
	Pressure	3	2	4		
	Other	4	3	4		

* *p* value for Chi-square test; ** *p* value for Mann–Whitney U Test.

**Table 4 ijerph-20-03468-t004:** Logistic regression model for the onset of sexual activity. Multivariable Regression Results. (SE—standard errors for coefficients; OR—odds ratio; 95% confidence intervals (CI) for the logistic regression model of the onset of sexual activity).

Independent Variable (Reference Category)	Coefficient	SE	*p*-Value	OR	95% CI for Population Odds Ratio
Constant term	0.41	0.12	0.001	1.51	1.19–1.92
Religiosity (Yes)	1.11	0.06	<0.0001	3.03	2.7–3.41
Drug use (No)	0.90	0.14	<0.0001	2.47	1.88–3.24
Living in student housing (Yes)	0.56	0.08	<0.0001	1.76	1.5–2.06
Talking with parents about contraception (No)	0.48	0.08	<0.0001	1.61	1.38–1.88
Living with parents (Yes)	0.29	0.07	<0.0001	1.33	1.17–1.51
Talking with parents about sex (No)	−0.19	0.08	0.014	0.82	0.7–0.96
Gender (F)	−0.24	0.07	<0.0001	0.78	0.69–0.89
Smoking (Yes)	−0.47	0.07	<0.0001	0.63	0.55–0.72
Alcohol consumption (Yes)	−0.470	0.07	<0.0001	0.63	0.55–0.72

**Table 5 ijerph-20-03468-t005:** Univariate Analysis of the of behaviour influences on the age of sexual initiation.

Characteristics		%/* y.o.	t	*p* Value
Have you ever smoked cigarettes?	Never	62%	−12.42	<0.0001
	Yes	38%		
Have you ever drunk alcohol?	Never	15%	−3.12	0.0018
	Yes	85%		
Have you ever used drugs?	Never	86%	−9.97	<0.0001
	Yes	14%		
Do you masturbate?	Never	12%	−6.61	<0.0001
	Only in the past	10%		
	Less than once a week	47%		
	More than once a week	31%		
Do you watch pornography?	Never	18%	−8.20	<0.0001
	Only in the past	16%		
	Less than once a week	49%		
	More than once a week	18%		
Place of residence	Village	25%	−7.11	<0.0001
	under 20 k inhabitants	12%		
	20 k−50 k inhabitants	12%		
	50–100 k inhabitants	11%		
	100–200 k inhabitants	10%		
	200–500 k inhabitants	13%		
	over 500 k inhabitants	18%		
At what age did you first watch a pornographic movie?		14.39 *	13.67	<0.0001
Do your religion beliefs influence your sex life?	No	73%	13.84	<0.0001
	Yes	27%		

* y.o.—years old.

**Table 6 ijerph-20-03468-t006:** Assignment Sheet of the Influencing Factors of the age of sexual initiation.

Characteristics	Assignment
Religiosity	0 = Religion has no impact on life decisions 1 = Religion has a significant impact on life decisions
Age of first pornography viewing	Initial data
Size of city of residence	1 = Village 2 = under 20 k inhabitants 3 = 20–50 k inhabitants 4 = 50–100 k inhabitants 5 = 100–200 k inhabitants 6 = 200–500 k inhabitants 7 = over 500 k inhabitants
Life conditions	1 = Poor 2 = Adequate 3 = Good 4 = Very good
Frequency of masturbation	1 = Never 2 = Only in the past 3 = Less than once a week 4 = More than once a week
Frequency of pornography viewing	1 = Never 2 = Only in the past 3 = Less than once a week 4 = More than once a week
Drug use	0 = Never 1 = Yes
Alcohol consumption	0 = Never 1 = Yes
Smoking	0 = Never 1 = Yes

**Table 7 ijerph-20-03468-t007:** Multivariable Regression Results of Factors Affecting the age of sexual initiation (SE—standard errors for coefficients).

Independent Variable	Coefficient	SE	t	*p* Value
Constant term	17.05	0.19	88.95	<0.0001
Religiosity	0.55	0.07	8.24	<0.0001
Age of first pornography viewing	0.11	0.01	11.61	<0.0001
Size of city of residence	−0.04	0.01	−3.44	0.0006
Life conditions	−0.10	0.04	−2.58	0.0099
Drug use	−0.32	0.08	−3.81	0.0002
Smoking	−0.55	0.06	−9.25	<0.0001

## Data Availability

Data supporting reported results are available on request from the study team.

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
