# Peer review of "Influence of Selected Sociodemographic and Cultural Factors on the Age of Sexual Initiation of Students from Polish Universities"

_ijerph, 2023, doi:10.3390/ijerph20043468_

Round 1

Reviewer 1 Report

This paper examines the effect of specific demographic and cultural factors on the age of sexual initiation in a large sample of Polish university students.

Though there are several studies examining this specific variable from both developed and developing countries, the current study appears to be the first of its kind from Poland, and the authors' work in this regard is to be appreciated.

There are certain aspects of the paper which would benefit from corrections or clarifications:

1. Introduction

a. The first paragraph, which discusses the general health benefits of sexual intercourse, appears to be disconnected from the rest of the introduction and from the study itself. As the current study only examines the age at first intercourse, this paper does not directly address issues related to the frequency of sexual intercourse and its long-term benefits. It would be preferable to delete this section and replace it with a general discussion of age at sexual initiation, its positive and negative aspects, and the variables influencing it.

b. The discussion on the predictors of the age at first intercourse requires some clarification. For example, the authors state that "race" is a predictor of age at first intercourse, but it is not clear if this association is significant after correcting for sociodemographic and cultural factors (e.g., family income, family structure / single parent families, peer pressure, different attitudes towards sexuality among different ethnic groups.) It is also not known if these results (which are from a U.S. study) apply to other multi-racial or multi-ethnic societies (e.g., Brazil or South Africa). As "race" refers to a genetic / biological construct, it would be better to reword this section in a more neutral way (e.g., "Research has shown that the age at sexual initiation differs significantly across ethnic groups in some multi-ethnic societies, and this may be due to X, Y, and Z...")

c. The authors have stated that access to "sexuality education" (sic) in Poland is limited. This statement is somewhat surprising as Poland is a high-income country with a high level of adult education and economic development. The authors should provide some support for this statement - e.g., the study by Izdebski et al. (2022) which included 595 young Polish adults and found that over 30% of them reported having no such classes in school. The authors could also discuss the cultural factors that could account for this unusual finding.

2. Aim

The operational definition of the study population is confusing. The authors state that they have defined "young adults" as those above the average age of sexual initiation in Poland; does this mean that they have used a different cut-off for men and women? (Lines 45-46 give different ages for each gender.) If so, why? It would surely be preferable to use a uniform cut-off for "young adults" (e.g. age 18-25 or even 18-26 as in the Izdebski et al. study).

3. Methods

a. The data for this study was collected in 2016, but the current paper has been submitted for publication in 2023. Is it possible that some of the study findings may have been superseded due to cultural or policy changes? What were the reasons (if any) for the lag between data collection and analysis / report writing?

b. In lines 95-96, it is sufficient to state that prior results from this survey have been published elsewhere; it is not necessary or customary to mention a specific journal (as the full citation is already in the reference list).

c. Apart from the variables studied by the authors, what other demographic or personal variables could have influenced sexual initiation? (e.g. family structure / single parent family, history of parental separation or divorce, peer pressure, role models from media / social media, income, history of childhood adversity such as physical or sexual abuse)? While I agree that it is not possible to cover every single factor in this study, these variables should at least be mentioned in the Discussion / Limitations if they could not be included in the current research.

c. How representative is the study sample of the general population? The authors have selected a sample of university students with a large female preponderance (approx. 70:30 ratio of women to men). Could a better sampling method have been used? What problems would the current sample pose when attempting to generalize these results, or use them to guide policy or educational approaches?

d. Given that the authors have examined multiple predictors of the age of sexual initiation, was a multivariate analysis (e.g. binary logistic regression of the factors differentiating those who have not initiated sexual activity vs those who have)? If not, why?

e. As multiple comparisons were made using the chi-square test, was a post-hoc correction (e.g. Bonferroni's method of p divided by number of comparisons) attempted? If not, could any of the findings represent "false positives"?

4. Results:

a. A typesetting instruction ("section may be divided...") appears to have been inadvertently included in line 172; this can be deleted.

b. What statistical test(s) were used for the analyses in all the tables, particularly Table 3? Ideally, both the test statistic (i.e., the chi-square or Mann-Whitney value) and the p-value should be mentioned.

c. The authors have referred to Spearman rank correlations in the Methods section; where are these results presented?

5. Discussion

This section is generally well-written, but it could be improved by modifying the tone of lines 269-272. While the authors' general conclusions may be correct (i.e., that religious and cultural values may have led to a conservative or "repressive" attitude towards sexuality and sex education in Poland), it is preferable to avoid a moralizing or editorializing tone ("It is disturbing..."). This section would be stronger if the authors presented facts to support this discrepancy (e.g., data from other European countries that portray a different attitude towards sexuality) and listed the possible contributing factors (religious beliefs, cultural atittudes towards gender and sex), thus allowing readers to draw their own conclusions.

On the whole, this paper is a valuable contribution to the literature and I look forward to reviewing a revised version of this manuscript.

Author Response

Dear Reviewer, 

Kind regards, 

Authors

Reviewer 2 Report

The topic this manuscript is interesting and timely. However, the manuscript has major methodological error in design and analysis. The title of the research does not match the content of the manuscript nor the data analyses.

The title and the text of refer to the Influence of Selected Sociodemographic and Cultural Factors 2 on the Age of Sexual Initiation of Students from Polish Universities. I do not understand how the authors find this information. The whole tables are only shows differences between males and females. How do you know the influences? Where are the tables that confirm this influence?

The authors needs to re-analyze the data by undertaking a regression analysis to find whether there is influence or not.

There are many parts in the manuscript that different arguments were assumed with no solid data or justification. For example, lines 25-27: Place of residence, type of housing, religion, and field of study influence the onset of sexual initiation. Gender, parents' education, and social conditions did not influence the age of sexual initiation. How do you know this?

Parts of the manuscript is contracting each other’s.   lines 20-21 is contracting the data in table 3.

I think the manuscript needs re-write.

Author Response

Dear Reviewer, 

Kind regards, 

Authors

Reviewer 3 Report

In the section "Materials and methods":

The type of study carried out should be indicated, since it is not mentioned.

The study has a significant selection bias, so it must be highlighted in the limitations, and it must be pointed out that the results must be treated with great caution

Author Response

Dear Reviewer, 

Kind regards, 

Authors

Reviewer 4 Report

The paper meets the Journal's aim and requirements. The main findings obtained in sociological research involve the essential pool of respondents. However, some important details should be highlighted as in the present form the results remain incomprehensive to some extent.

1) It is unclear why the authors conducted the review in 2016 but publish the results only in 2023. If some ethical or other obstacles have arisen for 7 years it should be explained in the section with the limitations for the research. In the given form the use of 2016 data seems to be outdated and doubtful for an explanation of the current behavioral changes. They can be driven by different newest factors, like Covid pandemic and others. In any case, the use of such an outdated dataset demands careful justification;

2) I doubt the representativeness of the results as some essential methodological facts are missed or unexplained: the confidence level, standard error and overall population size;

3) the gender distribution of the respondents is unequal and obviously irrelevant for the total population of the target group - see line 98;

4) the authors write about the sample involving students aged 18-35. It is doubtful that students aged 25 and over really took part in the survey. At least, the authors should highlight the age distribution in detail, for example, in Table 1. It is too generalized in the current version of the article;

5) the authors write about the questionnaire of 31 questions referring also to their previous work - [16]? However, if comparing the appendix and source 16, the questionnaires differ. In this manuscript, there is only 20 questions are mentioned;

6) the title of Table 2 is incorrect - only three questions are connected with drugs;

7) numerous technical gaps are typical for this work. First of all, I believe, the authors meant "sexual education" instead of "secular education" in the Abstract - see line 24. See also Table 2. There is unclear abbreviation (STD) - it should be explained; two last questions contain the option "Well"... it should be "No"? The sign "*" near the questions means that there should be notes under the Table, but they are absent.

8) Mentioning their previous work (line 96), the authors missed the numbers. Am I right and it should be 16? 

9) describing the type of residence place, the authors include villages and ... cities? - the word is missed, only size is available. See Table 1.

Author Response

Dear Reviewer, 

Kind regards, 

Authors

Round 2

Reviewer 1 Report

The revisions made by the authors are satisfactory in my opinion. I commend the authors for their thorough work and have no further major changes to suggest.

Reviewer 2 Report

Many thanks for considering my comments on the earlier version of the manuscript. The manuscript is much improved and becomes more acceptable for publication. Well done 

Reviewer 4 Report

Considering the changes fulfilled by the authors, the paper is recommended to be published. However, in your further research, please, try to investigate recent data. It really looks strange to use a dataset collected a few years ago to share the findings on nowadays reality.